# Pulmonary function analysis in cotton rats after respiratory syncytial virus infection

**Margaret E. Martinez** *, **Olivia E. Harder, Lucia E. Rosas, Lisa Joseph, Ian C. Davis, Stefan Niewiesk**

Department of Veterinary Biosciences, College of Veterinary Medicine, The Ohio State University, Columbus, Ohio, United States of America

* Martinez.610@osu.edu

**Data Availability Statement:** All relevant data are within the manuscript and its Supporting Information files.

## Abstract

The cotton rat (*Sigmodon hispidus*) is an excellent small animal model for human respiratory viral infections such as human respiratory syncytial virus (RSV) and human metapneumovirus (HMPV). These respiratory viral infections, as well as other pulmonary inflammatory diseases such as asthma, are associated with lung mechanic disturbances. So far, the pathophysiological effects of viral infection and allergy on cotton rat lungs have not been measured, although this information might be an important tool to determine the efficacy of vaccine and drug candidates. To characterize pulmonary function in the cotton rat, we established forced oscillation technique in uninfected, RSV infected and HDM sensitized cotton rats, and characterized pulmonary inflammation, mucus production, pulmonary edema, and oxygenation. There was a gender difference after RSV infection, with females demonstrating airway hyper-responsiveness while males did not. Female cotton rats 2dpi had a mild increase in pulmonary edema (wet: dry weight ratios). At day 4 post infection, female cotton rats demonstrated mild pulmonary inflammation, no increase in mucus production or reduction in oxygenation. Pulmonary function was not significantly impaired after RSV infection. In contrast, cotton rats sensitized to HDM demonstrated airway hyper-responsiveness with a significant increase in pulmonary inflammation, increase in baseline tissue damping, and a decrease in baseline pulmonary compliance. In summary, we established baseline data for forced oscillation technique and other respiratory measures in the cotton rat and used it to analyze respiratory diseases in cotton rats.

## Introduction

Respiratory syncytial virus (RSV) infection of infants and children younger than 5 years of age causes an estimated 200,000 fatalities worldwide each year [1]. Hospitalization is often due to severe bronchiolitis and obstructive airway disease that is attributed to bronchiolar (small airway) inflammation, edema, necrosis and intraluminal plugs comprised of sloughed necrotic epithelium, mucus, and inflammatory cells [2, 3]. Ultimately, the clinical manifestation of hospitalized infants from RSV infection is decreased pulmonary compliance, increased airway resistance, wheezing and hypoxemia with a mean arterial oxygen saturation ($S_aO_2$) of 85% [2,

**Funding:** M.E.M. was funded through the Genetech Fellowship under grant number GRT00044407 from Genentech (https://www.gene.com/). O.E.H. and S.N. were funded through grant number P01AI112524 from the National Institute of Health National Institute of Allergy and Infectious Diseases (https://www.niaid.nih.gov/). L.E.R., L. J. and I.C.D. were funded through grant number HL137090 from National Institutes of Health/National Heart Lung & Blood Institute (https://www.nhlbi.nih.gov/). The funders had no role in study design, data collection and analysis, decision to publish, or preparation of the manuscript.

**Competing interests:** Genentech provided the corresponding author with a research fellowship that aided in the following research to be conducted. This does not alter our adherence to PLOS ONE policies on sharing data and materials.

4, 5]. RSV bronchiolitis in infants has also been associated with recurrent wheezing, asthma, and pulmonary dysfunction such as airway hyper-responsiveness (AHR) later in childhood [2, 6]. However, there are disagreements in the literature on whether abnormal lung function predisposes infants to RSV infection or whether RSV infection causes abnormal lung function [5, 7].

In order to understand the pathophysiology of RSV infection, many laboratories have investigated the effects of RSV on the mouse lung with inconsistent results. A few studies reported baseline respiratory perturbations after RSV infection [8–10], while several others did not [11–22]. Some studies concluded that RSV in mice is associated with AHR during acute infection [10, 14, 16–19, 23, 24], particularly after re-infection or with confounding conditions like lung injury or allergen sensitization [11–15, 17, 20–22, 25]. However, other studies did not find a correlation between AHR and RSV infection [11, 12, 20–22, 24–26]. A contributing factor to divergent results might be the use of plethysmography or breath distention techniques which have a relatively low sensitivity [8–11, 14, 15, 17, 19–21, 23–25, 27–29]. Another confounding factor might be that mice are not an ideal model for RSV infection as they require a supraphysiologic inoculum for infection and the virus replicates poorly in the lung and does not replicate in the nasal epithelium [30, 31]

In contrast to the mouse, the cotton rat is susceptible to upper and lower respiratory tract infections with RSV, and requires a lower inoculum similar to that seen in infants for infection [31, 32]. In this study, we attempted to characterize pulmonary function in cotton rats after RSV infection using the most sensitive techniques available. We established a protocol for cotton rats which allowed us to use forced oscillation technique, via the flexiVent system, to accurately define the effects of asthma and RSV on lung mechanics (e.g. resistance and compliance) [28, 33, 34]. In addition, we measured arterial blood gases during infection to accurately quantify the clinical perturbations of lung function [35]. Furthermore, we quantified the impact of RSV on airway mucus production using a precise color deconvolution algorithm [9] and compare RSV infection with allergy sensitization after house dust mite antigen (HDM) injection.

## Material and methods

### Cotton rats

Inbred cotton rats (*Sigmodon hispidus*) were housed in polysulfone microisolation cages (NextGen Rat 900, Allentown Inc., Allentown, NJ, USA) in a barrier facility with a 12:12 h light cycle. Cotton rats were maintained in an environment with a temperature of $20 \pm 2°C$ and 30% to 70% relative humidity. Cotton rats were 6–8 weeks of age. All studies were approved by The Ohio State University Institutional Animal Care and Use Committee.

### Viruses

Stocks of RSV A2 and RSV line 19F were grown in human epithelial type 2 (HEp2) cells in MEM supplemented with 2% fetal calf serum as previously described [36]. Viruses were sucrose purified and titered on HEp2 cells.

### RSV infection

Animals were sedated with isoflurane administered in a chamber and subsequently inoculated intranasally (IN) with 100μl virus ($1x10^5$ $TCID_{50}$).

## Lung fixation and histologic examination of inflammation and mucus production

Cotton rats were euthanized by carbon dioxide inhalation. The lungs were inflated with 1mL of 4% paraformaldehyde (PFA) and fixed in 4% PFA. All tissues were routinely processed, embedded in paraffin wax, and sectioned (4μm) according to the revised guide for organ sampling and trimming in rats and mice by the European Experimental and Toxicologic Pathology group [37].

Cotton rat lung histology slides were stained routinely with hematoxylin and eosin (H&E). Light microscopic evaluation was performed by a veterinary anatomic pathologist (MM) (model CX41, Olympus, B and B Microscopes Limited, Pittsburgh, PA, USA). A semi-quantitative scoring system, adapted from previous studies [9] was used (S1 Table). The sum of the mean scores were calculated to produce a total histologic inflammatory score.

For detection of mucus production, cotton rat lung histology slides were stained with Periodic acid-Schiff (PAS) and Alcian Blue. Whole slide images were scanned at 40x magnification (ScanScope XT, Aperio Technologies). Similar to previous studies, quantification of mucus staining was performed using the Color Deconvolution algorithm available in the Image Analysis Toolbox (Aperio Technologies) [9, 13]. The total percent positive staining cells, percent strong positive staining, and average strong staining intensity were calculated and compared between groups. A histologic semi-quantitative scoring system based on the percentage of PAS positive bronchiolar cells present was applied (S2 Table).

## Broncho-alveolar lavage

Broncho-alveolar lavage fluid (BALF) was collected as previously described [36]. BALF was collected from cotton rats immediately after euthanasia. The trachea was cannulated, and the lungs were lavaged with 1mL of PBS supplemented with 1% protease-free BSA. The BALF was kept on ice until processed (Comparative Pathology and Mouse Phenotyping Shared Resource, Ohio State University). Automated nucleated cell counts were performed by using a Forcyte veterinary hematology analyzer (Oxford Science, Oxford, United Kingdom).

## Lung wet: Dry weight ratio

Lung wet: dry weight ratio was measured as previously described [38, 39]. Rats were euthanized and their lungs were removed, weighed, and dried in an oven at 55˚C for 48 hours. After drying, lungs were weighed again. Wet: dry weight ratio was then calculated as an index of intrapulmonary fluid accumulation, without correction for blood content.

## Pulse oximetry measurements

The medial thighs and ventral neck were clipped 48 hours before measurements were taken. Cotton rats were sedated by inhaled isoflurane initially in a chamber, and then maintained with a nose cone on room air ($F_iO_2 = 0.21$). Peripheral capillary oxygen saturation ($S_pO_2$), heart rate, pulse distention, and breath rate were measured with the collar clip sensor (Starr Life Sciences Corp., Allison Park, PA, USA) in accordance with manufacturer's instructions. The same parameters as well as breath distention were measured with the thigh clip sensor. Data were collected for a minimum of 10 seconds (150 data points) per animal [39].

## Measurement of arterial blood gases

Cotton rats were anesthetized as for flexiVent studies, tracheotomized, and mechanically ventilated on room air ($F_iO_2 = 0.21$). No other lung function tests were performed before or after blood collection, as this was a terminal procedure. The carotid artery was isolated and ligated

cranially in order for 200μL of blood to be collected in a heparinized 1mL syringe. $pH_a$, $P_aCO_2$, and $P_aO_2$ were measured using ab *EG6+* cartridge in an iSTAT® blood gas analyzer (Abbott Laboratories, Abbott Park, IL, USA). The P/F ratios were calculated by dividing the arterial partial pressure ($P_aO_2$) by the fraction of inspired oxygen ($F_iO_2$), which on room air is 0.21.

## House dust mite (HDM) antigen sensitization

100μg house dust mite (HDM, *Dermatophagoides pteronyssinus*) antigen was absorbed in aluminum phosphate (AdjuPhos, Brenntag, Ballerup, Denmark) at 1:1 ratio for 30 minutes at room temperature. HDM was administered intraperitoneally (IP). Sensitization was followed 8 days later with intranasal (IN) administration of 100μg HDM in a volume of 100μL PBS. Four days after IN HDM administration, cotton rats were anesthetized and forced oscillation technique was performed or lungs were collected for histologic examination.

## Anesthesia and tracheotomy protocols

Cotton rats were first sedated with isoflurane administered in a chamber for 7 minutes. Rats were next administered 10mg/kg diazepam IP, followed by 100mg/kg ketamine IP 7 minutes later. A surgical plane of anesthesia was determined by the lack of a response to interdigital forceps grasp. If a surgical plane was not achieved within 15 minutes of ketamine dosage, an additional half dose of ketamine was administered.

Seventy percent ethanol was applied to the ventral neck, and the ventral neck skin was excised. The salivary glands were bluntly separated and placed laterally, and the sternohyoid and sternothyroid muscles were removed exposing the trachea. A scalpel was used to incise the trachea between the first and second tracheal rings, and a 4cm 15 gauge gastric tube (cat. 724446, Harvard Apparatus, Holliston, MA, USA) was placed as the tracheal cannula. A circumferential suture was tightened around the exterior of the trachea.

## Forced oscillation technique

Mechanical properties of the cotton rat lung were assessed using forced oscillation technique, which is a terminal procedure. Anesthesia and tracheotomy were performed as described above. Rocuronium was administered (10mg/mL IP), and the rat was ventilated on a computer-controlled piston ventilator (flexiVent, SCIREQ, Montreal, Quebec, Canada) with a tidal volume of 10mL/kg at a frequency of 90 or 150 breaths/minute against $3cmH_2O$ positive end-expiratory pressure (PEEP). Both the flexiVent Module 2 (4.8mL stroke FV-M2) and the FX module 3 were used with the previous settings. Following two total lung capacity (TLC) maneuvers, pressure and flow were collected during a series of standardized volume perturbation maneuvers. Total lung resistance (R) and elastance (E) were calculated using the single-compartment model and used to derive further parameters of respiratory function: $R_N$ (Newtonian resistance, composed mostly of the flow resistance of the conducting pulmonary airways), G (tissue damping, reflective of resistance of peripheral airways and parenchyma), and H (parenchymal tissue elastance). The flexiVent calibration procedure removes cannula impedance from the reported data. Residual inertance (I) is therefore negligible and is not reported herein [26, 40]. If the cannula contained fluid on extubation or the subject died before completion of the lung function measurements, then that subject was excluded from the study.

## Methacholine challenge

Following baseline recordings of lung-function parameters with saline-only nebulization, airway hyper-responsiveness was assessed on the flexiVent exposed to increasing doses of

methacholine (0.1, 1.0, 10.0, 20.0, and 50.0 mg/ml) in sterile normal saline prepared fresh daily. Each methacholine dose was delivered over a 10 second period via an AeroNeb vibrating plate ultrasonic nebulizer, in series with the inspiratory limb of the flexiVent Y-tube. Ten recordings of R and E were generated following administration of each methacholine dose, and mean values of all 10 measurements for each parameter at each methacholine dose were thereby obtained [26, 40].

## Statistics

Statistics were performed using GraphPad Prism Software version 6.07 (San Diego, CA, USA). Differences between groups were analyzed using a One-Way ANOVA or unpaired two-tailed student T test, with Tukey's multiple comparison posttests. P < 0.05 was considered statistically significant. Individual experiments were repeated 3 to 5 times. Aggregate animal numbers were statistically analyzed. Males and females were used for each parameter measured and compared to each other to determine the effect of gender. All data are represented as means +/- standard deviations with individual data points.

## Results

### RSV infection causes a mild increase in pulmonary inflammation but not an increase in mucus production

In order to evaluate lung pathophysiology of cotton rats after RSV infection, we assessed histology, mucus production, tissue edema, blood oxygenation and lung mechanics. The first step of the study was the characterization of the inflammatory response to RSV in the cotton rat lung so that the histologic findings could be compared to the lung function analysis and to allergen sensitization with house dust mite (HDM) antigen [15, 36]. Lungs were assessed histologically for inflammation based on the number of peribronchiolar, perivascular, bronchiolar, interstitial and alveolar inflammatory infiltrates and assigned a score of 0–4 based on a scoring scheme as described in S1 Table [9]. After RSV infection, peak titers of viral replication are observed on day 4 and 5, with a virus-specific immune response being detectable on day 5. Consistent with these findings was the moderate increase in inflammatory infiltrates 4 and 5 days post-infection (DPI) with RSV A2 (Fig 1A) with a mean histologic inflammatory score on day 4 of 4.3 +/- 1.9 and on day 5 of 5.3 +/- 0.9 (out of 30). This correlated with a low number of white blood cells obtained by broncho-alveolar lavage (BAL) (Fig 1B). The type and location of inflammatory infiltrates was similar between 4 and 5 days post-RSV infection (Fig 1C). HDM-sensitized cotton rats demonstrated a statistically significant higher histologic inflammatory score of 13.2 +/- 1.2 (Fig 1A) which correlated well with the number of cells within fluid obtained by broncho-alveolar lavage (BALF). Uninfected cotton rats had an average white blood cell count (WBC) of 444 +/- 113.5/ul, while 5 DPI RSV cotton rats had a mean WBC/ μL in BALF of 405 +/- 112, and HDM sensitized cotton rats had a mean WBC/ μL in BALF of 1025 +/- 188.

### Mucus production after RSV infection

The strong mucus production observed in infants after infection with RSV can lead to disturbances in lung mechanics (increased baseline airway resistance). To obtain comparative data in cotton rats, we quantified bronchiolar mucus production after infection with two strains of RSV. Whereas RSV A2 is a commonly used laboratory strain, RSV line 19F is a recombinant RSV which expresses a fusion protein associated with increased (two-fold from a very low background) mucus production in the mouse [8, 16, 41]. Infection with

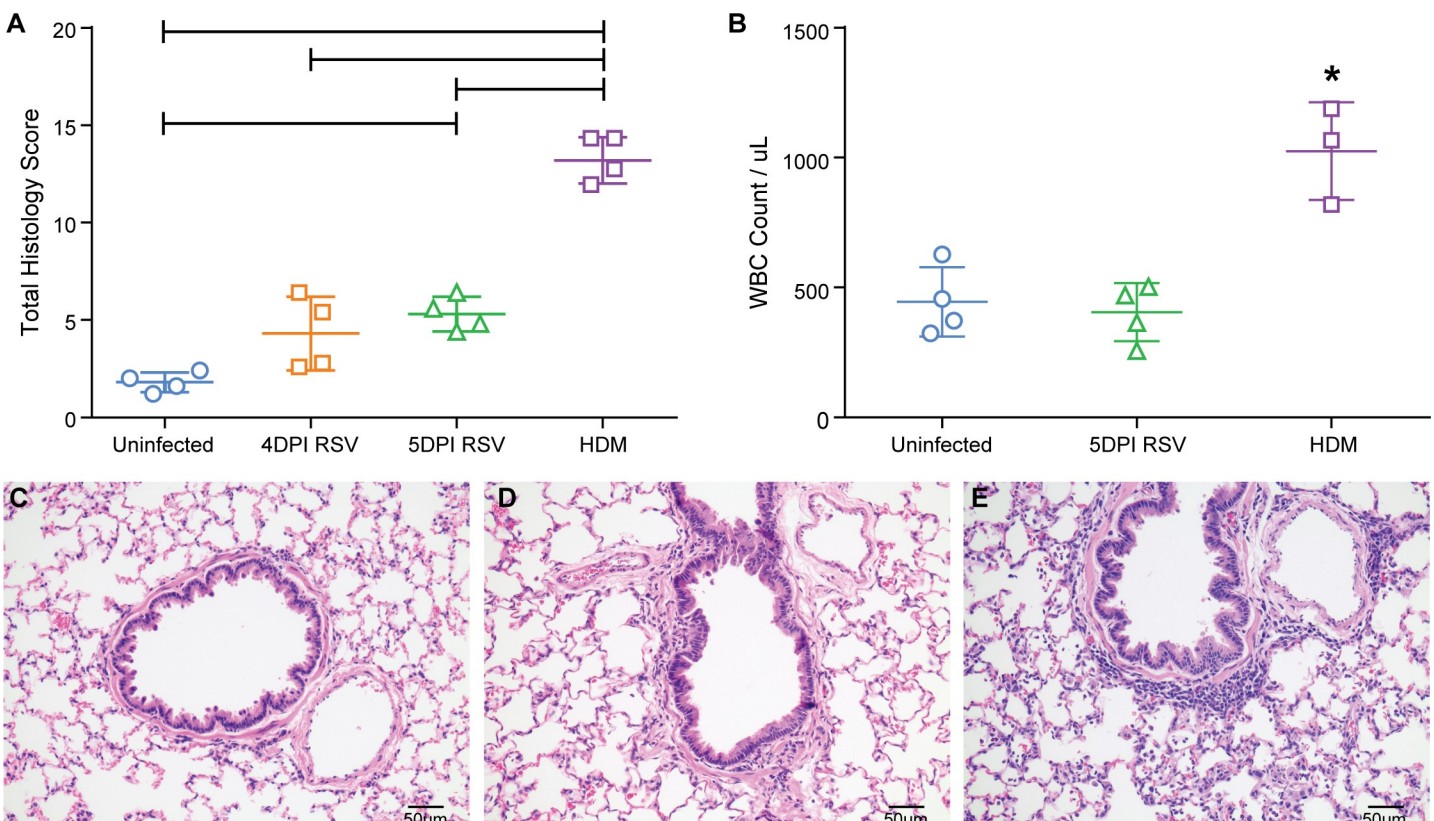

**Fig 1. Inflammation after RSV A2 infection.** 1a. The semi-quantitative scoring of inflammation in lung sections of uninfected cotton rats, cotton rats 5 and 4 days post-RSV infection, and HDM-sensitized cotton rats was compared. The mean and standard deviations are represented (n = 4/ group). Asterisks indicate a p value < 0.05 by One-Way ANOVA. 1b. The total white blood cells per μL of bronchoalveolar lavage fluid (WBC/BALF) were compared between uninfected cotton rats, 5 days post-RSV infected cotton rats, or HDM sensitized cotton rats. Individual points represent the mean WBC/BALF from one animal with the mean and standard deviations represented too; 19DPI AAV-GFP (n = 3), 19DPI AAV-G (n = 3), 5DPI RSV (n = 4). Astericks indicate a p value < 0.05 by One-Way ANOVA. 1c. An H&E stain of an uninfected cotton rat lung representative of no to minimal inflammation. 1d. An H&E stain of a 4dpiRSV cotton rat lung representative of mild inflammation characterized by small numbers of peribronchiolar and perivascular lymphocytes and histiocytes. 1e. An H&E stain of a HDM sensitized cotton rat lung representative of moderate inflammation characterized by many peribronchiolar and perivascular lymphocytes and histiocytes, and numerous granulocytes within alveolar septae (interstitial infiltrates), abundant macrophages and granulocytes within alveoli, and many granulocytes within bronchiolar epithelium.

both strains was evaluated for induction of mucus in the cotton rat, as previous mouse studies have suggested that RSV line 19F induces more mucus production than RSV A2 [16]. In tissue sections stained for mucus with Periodic acid-Schiff and Alcian blue, an average of 101 bronchioles per slide were analyzed using the color deconvolution algorithm in the Image Analysis Toolbox. The percentage of positive mucus staining (Fig 2), the percentage of strong positive mucus staining, and the average staining intensity were quantified. There were no significant differences in mucus staining of cells between uninfected, RSV A2 infected, and RSV line 19F infected cotton rats (Fig 2). In addition, there was no significant difference in the intensity of staining (mean percent of strong positive staining cells) or the overall mucus production (mean staining intensity) between groups. When mucus production was evaluated 5 days post-infection using a semi-quantitative scoring system (S2 Table), there was still no significant difference in mucus production between the RSV A2 infected and uninfected cotton rats (S1 Fig). Therefore, RSV does not induce increased mucus production in the cotton rat.

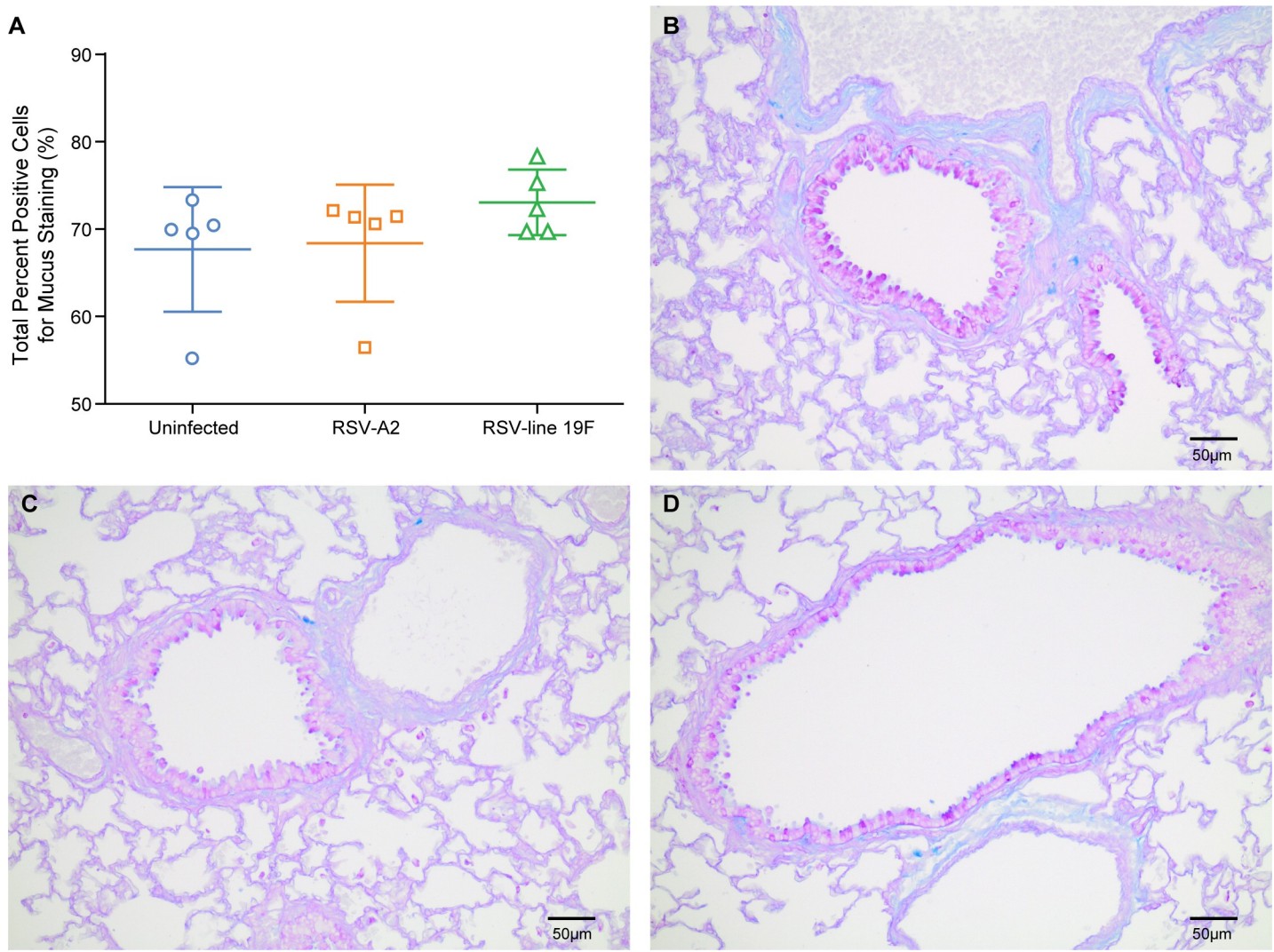

**Fig 2. Mucus production in cotton rats 8 days post-infection with RSV A2 and RSV line 19F.** 2a. Quantification of PAS and Alcian blue positive (mucus) staining in bronchioles was performed using the Color Deconvolution algorithm in Image Analysis Toolbox. Mucus production was assessed at 8dpi based on previous studies in the mouse [8, 16]. The mean total percent positive staining cells and standard deviations are represented (n = 5/ group). Asterisks indicate p value < 0.05 when compared to uninfected cotton rats by One-Way ANOVA. 2b. A representative airway and adjacent alveoli of a PAS/ Alcian blue stained section of uninfected cotton rat lung with little to no mucus production. 2c. A representative airway and adjacent alveoli of a PAS/ Alcian blue stained section of cotton rat lungs 8dpi RSV A2 infection with little to no mucus production. 2d. A representative airway and adjacent alveoli of a PAS/ Alcian blue stained section of cotton rat lungs 8dpi RSV line 19F infection with little to no mucus production.

### Increase in pulmonary edema has no effect on oxygenation

Another parameter of respiratory function is the development of pulmonary edema which decreases pulmonary compliance. The quantification of cotton rat lungs after RSV infection demonstrated a small but significant increase in the lung wet: dry weight ratios at 2 days post-infection with RSV compared to uninfected cotton rats (Fig 3). When only female cotton rats with corresponding lung function data were evaluated, there was no significant difference in pulmonary edema between the groups (S2 Fig).

To determine whether the increase in wet: dry weight ratios at day 2 post-infection altered gas exchange, we performed pulse oximetry to measure carotid and femoral oxygen saturation, heart rate, breath rate and breath distention. Collar sensor clips were applied to the clipped

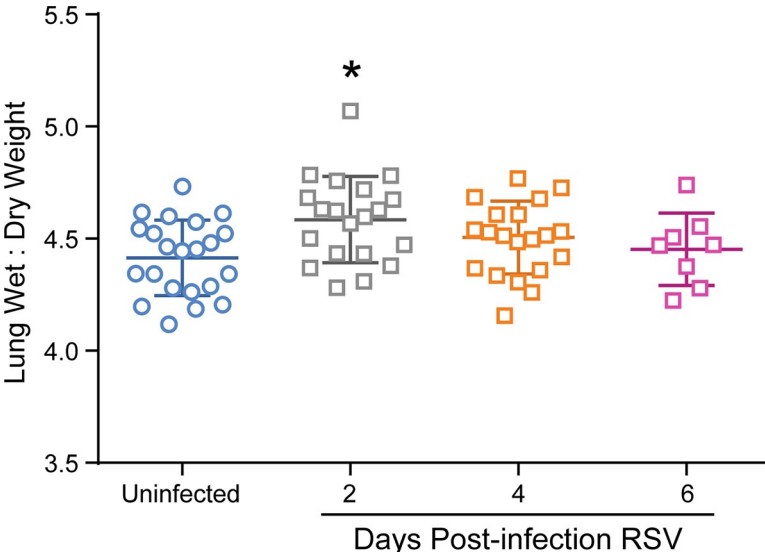

**Fig 3. Pulmonary edema after RSV A2 infection.** The lung wet: dry weigh ratios in uninfected cotton rats and cotton rats 2, 4 or 6 days post-RSV infection. Mean wet: dry weight ratios and standard deviations are represented. Asterisks indicate p value < 0.05 when compared to uninfected cotton rats by One-Way ANOVA; uninfected (n = 22), 2DPI RSV (n = 20), 4DPI RSV (n = 20), 6 DPI RSV (n = 8).

ventral necks over the carotid arteries. Thigh clips were applied to clipped medial thighs over the femoral artery. The thigh clip sensor also measured breath distention. Those parameters for uninfected cotton rats as well as cotton rats 2, 4, and 6 days post-RSV infection are in S3 and S4 Tables. The average peripheral oxygen saturation in uninfected cotton rats was 99.05% +/- 0.4, with an average heart rate of 525 +/- 18.5 beats per minute. On day 4 post-RSV infection, cotton rats did not have significantly higher mean heart rates and breath rates using femoral artery measurements. There was also no significant differences between $S_pO_2$, heart rate, and breath rate when the two clip locations were compared to each other. The only statistically significant difference was between the collar clip and thigh clip was pulse distention. This difference is expected and represents the variations in hydrostatic pressures between the carotid and femoral artery based on proximity to the heart.

Since oxygen saturations are a relatively insensitive measure of alveolar gas exchange, we also measured arterial blood gases from blood drawn from the carotid artery (S5 Table). Uninfected cotton rats had mean $P_aO_2$ values of 143 +/- 39, $P_aCO_2$ values of 26 +/- 9, $pH_a$ of 7.3 +/- 0.2, and P/F ratio of 679 +/- 186. Though day 2 and 4 RSV infected cotton rats had a lower mean $P_aO_2$ and P/F ratio, and higher $P_aCO_2$, all arterial blood gas parameters for the groups were not significantly different. In conclusion, there was no significant difference in blood oxygenation in cotton rats after RSV infection.

## Measuring lung function with forced oscillation technique in uninfected cotton rats

An increase in baseline airway resistance and decrease in pulmonary compliance has been documented in RSV hospitalized infants and children [5]. RSV infection is also associated with recurrent wheezing, asthma, and pulmonary dysfunction such as airway hyper-responsiveness (AHR) later in childhood [2, 6]. AHR is defined as the exaggerated narrowing of airways to a stimulus that does not result in comparable airway narrowing in healthy subjects [42–44]. In normal physiologic airway constriction, the stimulus for bronchoconstriction is acetylcholine,

which is released from peribronchiolar nerves. Therefore, AHR is assessed experimentally by comparing resistance between uninfected and infected subjects after stimulation with nebulized methacholine, an acetylcholine agonist.

The forced oscillation technique (FOT) is the preferred method for measuring lung function in children [28, 33, 34, 42] as it allows detailed and comprehensive assessment of lung mechanics. The most pertinent measurements are baseline and peak total airway resistance, baseline and peak tissue damping, and baseline pulmonary compliance as these are the lung function parameters that have been demonstrated to be altered in infected infants and children [5, 7]. FOT consists of applying an oscillatory waveform to a subject and measuring the pressure, flow, and volume responses to measure total airway resistance (R), pulmonary compliance (C), Newtonian resistance (large/ central airway resistance), tissue damping (G, peripheral/ small airway and alveolar resistance), and tissue elastance (alveolar elasticity/ the reciprocal of compliance). With these parameters, AHR is measured through increasing dosages of methacholine up to 50mg/mL.

In order to compare the effects of RSV infection on lung mechanics in cotton rats, we established reference data for uninfected animals using the FV-M2 flexiVent system (Fig 4). Uninfected male cotton rats a mean baseline airway resistance ($R_b$) of $0.22 cmH_2O.s/mL$ +/- 0.014, and following exposure to 50mg/mL methacholine had mean airway resistance ($R_{max}$) increased to $0.63 cmH_2O.s/mL$ +/- 0.16 (191.5% change from baseline +/- 85). Uninfected female cotton rats had a mean $R_b$ of $0.22 cmH_2O.s/mL$ +/- 0.018, and a mean $R_{max}$ of $0.72 cmH_2O.s/mL$ +/- 0.13 (226.7% change from baseline +/- 55). Our data demonstrated that the cotton rat does respond to methacholine exposure in an expected exponential fashion. There is an initial exponential increase in airway resistance after methacholine exposure, with eventual plateauing of airway resistance at higher dosages due to a presumed saturation of muscarinic receptors in the peribronchiolar smooth muscles.

## Female, and not male, cotton rats demonstrated perturbations in respiratory function measurements at 4 days after RSV infection

Several pulmonary function studies analyze respiration perturbations using only one gender [24, 26, 43]; however, one study found that there are gender differences to lipopolysaccharide induced AHR [44]. In patients, male children have a higher incidence and disease severity associated with RSV infection than female children [45]. Therefore, we analyzed the differences in airway resistance, tissue damping and pulmonary compliance in RSV infected female and male cotton rats. We compared pulmonary function at 4 days post-RSV infection since this is the time point in which there is peak viral replication in the lungs of cotton rats [46].

Four days after RSV infection male cotton rats had an $R_b$ of $0.25 cmH_2O.s/mL$ +/- 0.034, and a mean $R_{max}$ of $0.55 cmH_2O.s/mL$ +/- 0.13 (126.3% change from baseline +/- 71). Four days after RSV infection, female cotton rats had a mean $R_b$ of $0.26 cmH_2O.s/mL$ +/- 0.029, and $R_{max}$ of $0.9326 cmH_2O.s/mL$ +/- 0.19 (259% change from baseline +/- 73). There was a significant difference in $R_b$ and $C_b$ between uninfected and 4 days post-RSV infected female cotton rats, which was not seen in males (4A and 4E). There was also a significant difference between the $R_{max}$ (p = 0.0314) and $G_{max}$ (p = 0.0033) values of male and female 4 days post-RSV infected rats (Fig 4B and 4D). The $R_{max}$ values of 4DPI RSV infected male cotton rats were directly compared to uninfected male cotton rats using an unpaired two tailed student t-test, as well as 4DPI RSV infected female cotton rats to uninfected female cotton rats. There was a significant difference in $R_{max}$ between 4DPI RSV and uninfected female cotton rats (p = 0.0314), though not in 4DPI RSV and uninfected male cotton rats (p = 0.417). Similarly, there was a significant difference in $G_b$ when 4DPI RSV infected female cotton rats were

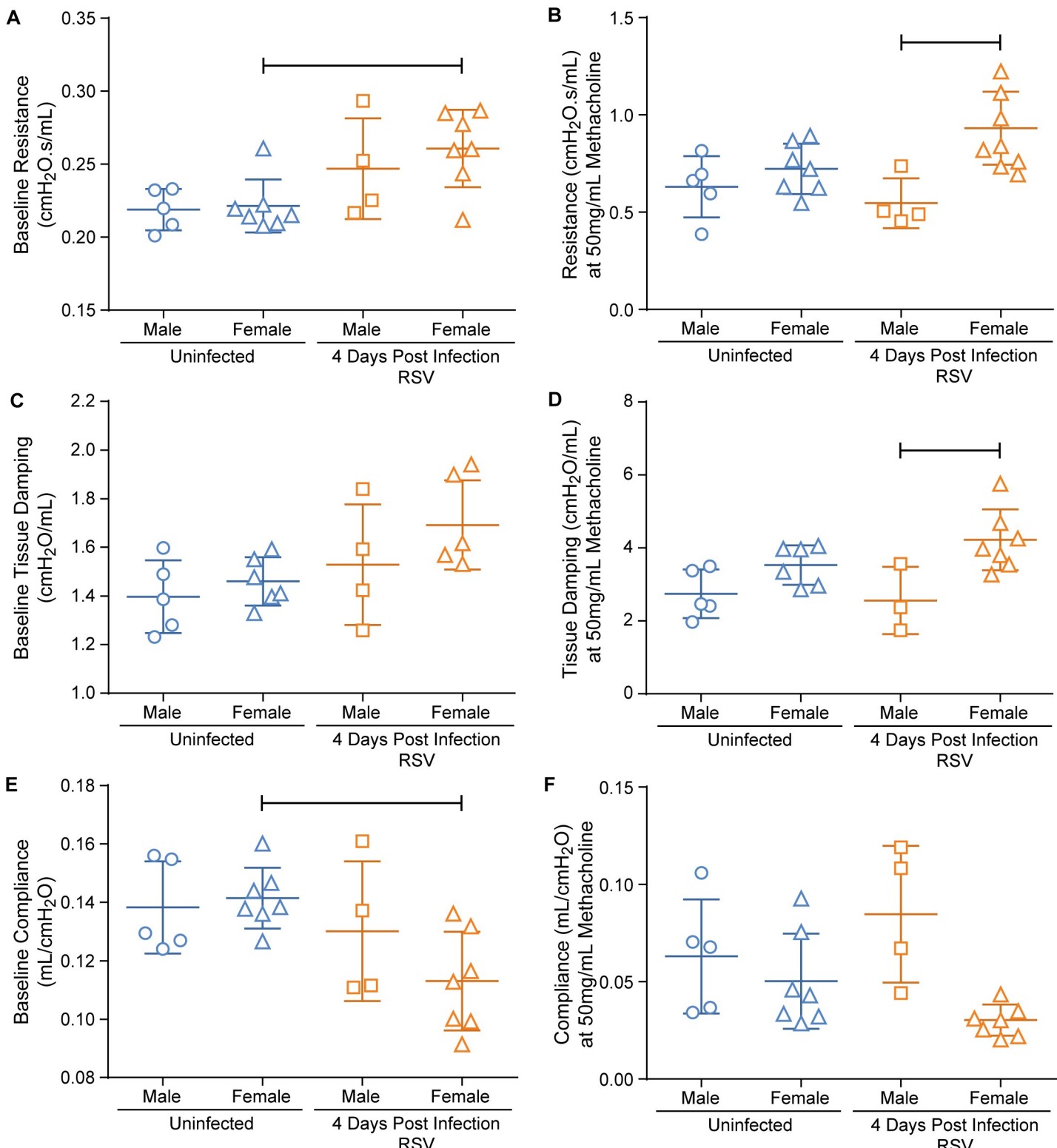

**Fig 4. Influence of gender on lung mechanics in uninfected and 4 days post-RSV A2 infection.** The mean and standard deviation for each group are represented. Fig a-b represent total airway resistance (R). Fig c-d represent tissue damping (G). Fig e-f represent dynamic pulmonary compliance (C). Fig a, c, and e represent the baseline measurements for R, G, and C. Fig b, d, and f represent the measurements obtained at 50mg/mL methacholine dosage for R, G, and C. Bars indicate $p < 0.05$ by One-way ANOVA; male uninfected (n = 5), female uninfected (n ranges 7–6), male 4DPI RSV (n ranges 4–3), female 4DPI RSV (n ranges 7–6).

compared to uninfected females using an unpaired two tailed student t-test (p = 0.0217). Therefore, when directly compared to their respective gendered uninfected rats, female cotton rats demonstrated AHR 4 days post-RSV infection while male cotton rats did not.

One possibility for the putative difference in lung function between the genders was a difference in airway lumen. Therefore, the mean airway luminal diameter of female and male cotton rats was measured using histologic sections. The mean airway diameter in male cotton rats was 155μm (+/- 25) and the mean diameter in females was 138μm (+/- 4), which were not significantly different from each other (S3 Fig). Also, there was no significant difference between uninfected male and female $R_b$ and $G_b$ to indicate one gender's airways being smaller than another (Fig 4A and 4C). Thus, it was determined that greater lung function disturbances observed in RSV infected female cotton rats were not due to a difference in airway morphology.

## RSV infection in cotton rats does not cause clinically significant respiratory dysfunction

To further evaluate lung function, cotton rats were intranasally infected with RSV and lung mechanics were measured on day 2, 4, 6, and 28 post-infection (Fig 5 and S4 Fig). When both genders and all time-points were compared, $C_b$ was significantly lower in 4DPI RSV cotton rats than uninfected cotton rats (S5 Fig). Similarly when females were analyzed separately without the male data, $C_b$ remained significantly lower at 4DPI RSV compared to uninfected (Fig 5E). The decrease in baseline compliance represents stiffer pulmonary parenchyma, which is an expected lung function disturbance in infants infected with RSV [5]. However, there was no significant difference in any other baseline parameters measured in any group compared to uninfected cotton rats (Fig 5A and 5C) as would also be expected to be different as seen in infected infants [5, 6, 47]. Lastly, there was also no evidence of AHR (Fig 5B and 5D). The lack of difference between these results comparing multiple groups versus comparing infected with uninfected cotton rats in the previous experiment might be due to the fact that biological differences have to be larger to be significant in multi-group comparisons or is due to instrumental sensitivity.

In order to determine whether the observed differences were due to instrumentation or statistical analysis, we compared lung mechanics in female cotton rats 4 days post-infection using the latest most sophisticated FX3 flexiVent module (SCIREQ, Montreal, Quebec, Canada). There were no significant differences in any of the parameters evaluated between uninfected and 4 days post-RSV infected female cotton rats (S6 Fig). For most of the parameters measured, the FX3 flexiVent system had smaller standard deviations compared to the FV-M2 flexiVent system.

## HDM sensitized female cotton rats demonstrate airway hyper-responsiveness, increased baseline tissue damping, and decreased pulmonary compliance

We applied our newly developed technology in cotton rats for the analysis of lung function to an allergy model (immunization with HDM). Cotton rats were sensitized with HDM and responses to increasing dosages of methacholine (0.1–50mg/mL) were measured (Fig 6; S7 Fig). HDM sensitization had no effect on $R_b$, with a mean of 0.26cmH$_2$O.s/mL +/- 0.040 (Fig 6A). However, following exposure to 50mg/mL methacholine, uninfected female cotton rats had mean $R_{max}$ increase to 0.72cmH$_2$O.s/mL +/- 0.13 (226.7% change from baseline +/- 55), whereas the mean $R_{max}$ of HDM sensitized rats was 1.15cmH$_2$O.s/mL +/- 0.13 (353.4% change from baseline +/- 80). There was a significant difference between the two groups $R_{max}$

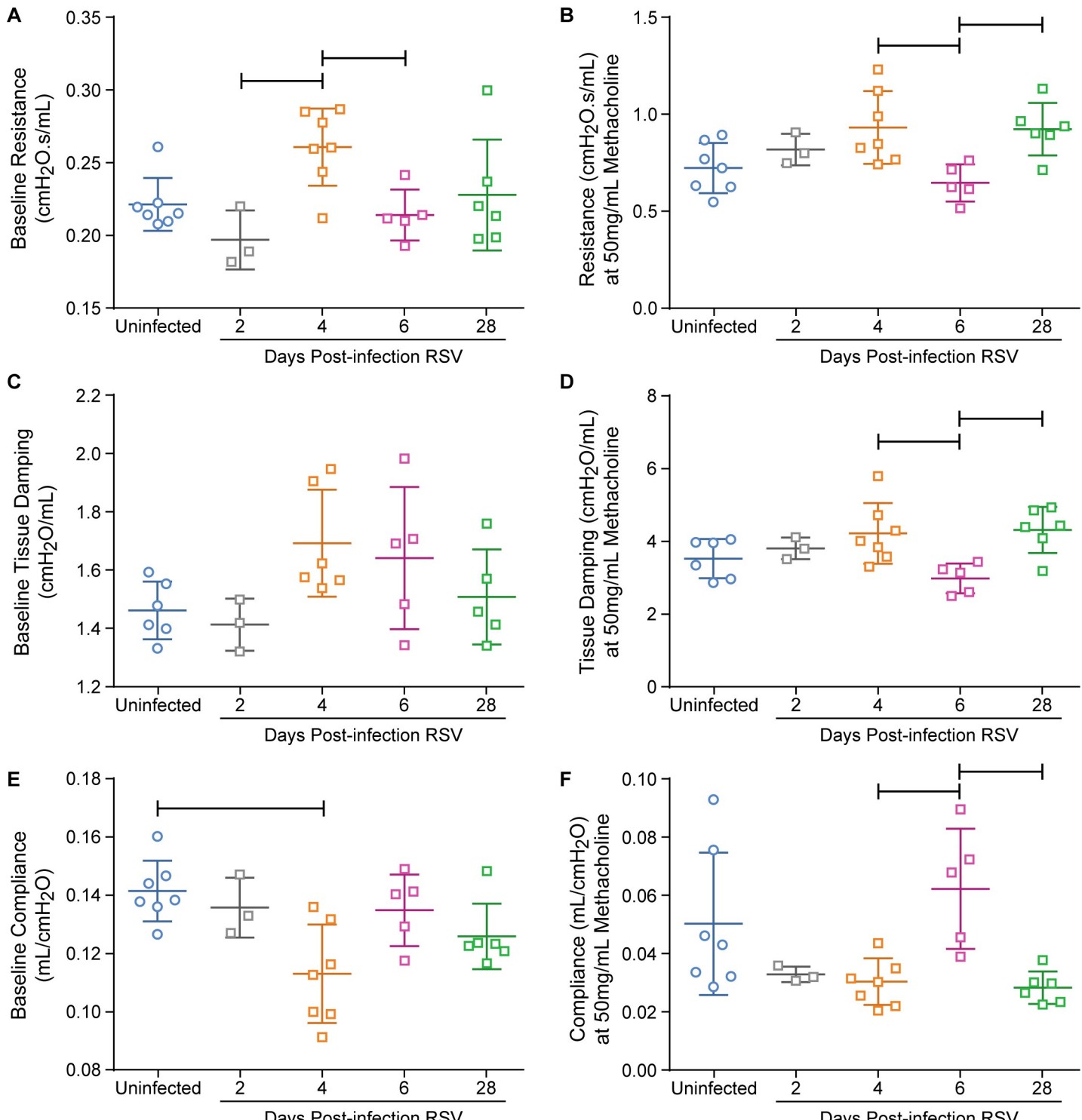

**Fig 5. Lung mechanics in uninfected cotton rats and 2, 4, 6 and 28 days post-RSV A2 infection female cotton rats.** The mean and standard deviation for each group are represented. Fig a-b represent total airway resistance (R). Fig c-d represent tissue damping (G). Fig e-f represent dynamic pulmonary compliance (C). Fig a, c, and e represent the baseline measurements for R, G, and C. Fig b, d, and f represent the measurements obtained at 50mg/mL methacholine dosage for R, G, and C. Bars indicate $p < 0.05$ by One-way ANOVA; uninfected (n ranges 7–6), 2DPI RSV (n = 3), 4DPI RSV (n ranges 7–6), 6DPI RSV (n = 5), 28DPI RSV (n ranges 6–5).

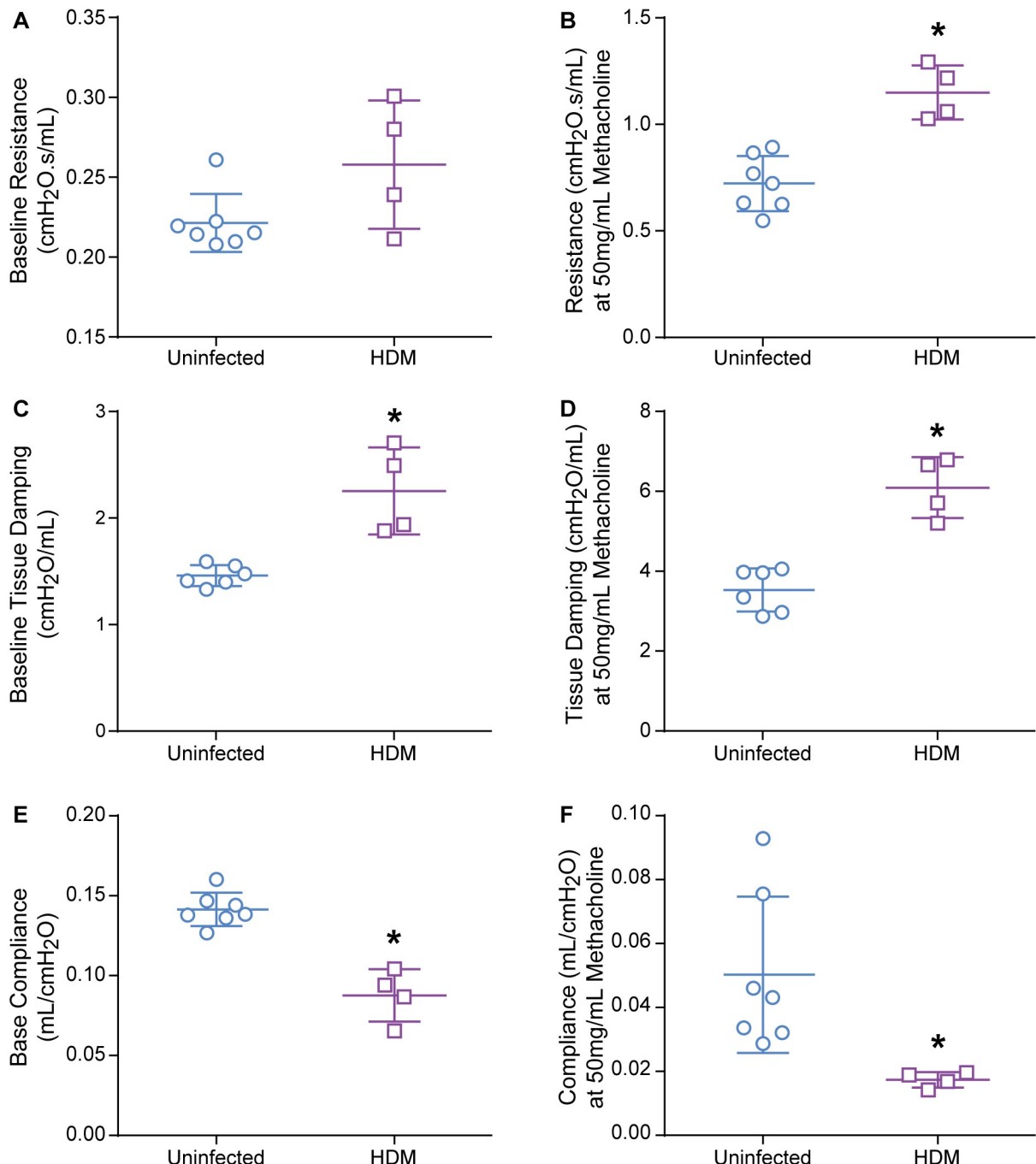

**Fig 6. Lung mechanics in uninfected cotton rats and HDM-sensitized female cotton rats.** The mean and standard deviation for each group are represented. Fig a-b represent total airway resistance (R). Fig c-d represent tissue damping (G). Fig e-f represent dynamic pulmonary compliance (C). Fig a, c, and e represent the baseline measurements for R, G, and C. Fig b, d, and f represent the measurements obtained at 50mg/mL methacholine dosage for R, G, and C. Asterisks indicate $p < 0.05$ by two-tailed unpaired Student T-test; uninfected (n ranges 7–6), HDM (n = 4).

(unpaired student t-test, p = 0.0005), as expected and therefore, the HDM sensitized female cotton rats demonstrated AHR (Fig 6B). Furthermore, there was a significant difference in baseline tissue damping ($G_b$) (p = 0.0016) as well as the tissue damping at 50mg/mL

methacholine dosage ($G_{max}$) (p = 0.0002) between the two groups (Fig 6C and 6D). The difference in tissue damping supports the interpretation of AHR as well as localizing the airway hyper-responsiveness to the smaller airways. Although $R_b$ was not different from uninfected rats, $G_b$ was significantly higher in HDM sensitized rats indicating small airway obstruction (Fig 6C). There was also a significant difference (p<0.0001) in baseline pulmonary compliance ($C_b$) as well as compliance ($C_{max}$) measured at 50mg/mL methacholine dosage (p = 0.277) in HDM sensitized cotton rats compared to uninfected female cotton rats (Fig 6E and 6F).

## Discussion

This is the first report of forced oscillation technique (FOT) in the cotton rat, which allows for detailed and precise characterization of lung mechanics in various conditions like infections and allergic disease. The cotton rat is susceptible to several respiratory viruses, such as RSV and HMPV, and is considered an excellent small animal model for RSV research [31, 32]. In order to characterize the pathophysiology of lung infection, we set out to establish sensitive pulmonary function measurement techniques. Our results provide baseline data for lung physiology studies in the cotton rat. We found that in contrast to mice [26, 40, 42, 43], cotton rats were highly sensitive to ketamine, and thus accurate dosing of ketamine based on weight as well as a quiet, low stress environment were required. A technical challenge was the sensitivity of a vagal response to slight manipulation of the peritracheal vagal nerve during tracheotomy.

Our data indicate differences in measurements between the FV-M2 and FX3 flexivent systems, a difference that has also been observed by other groups [48]. One study found that there was no difference in precision or accuracy in resistance between the models; however, the FX system resulted in more accurate compliance measurements [48]. Therefore, the difference in significance in the parameters measured in 4DPI RSV infected female cotton rats between the different instruments may have been partly attributed due to their varying configuration.

Another difficulty in assessing the effect of RSV infection on lung function, is the question of how many groups should be included in the analysis [9, 12, 16, 18, 19, 26, 41, 49, 50]. Whereas small differences in lung function are statistically significant in pairwise comparisons, they are not sufficient to be detected at a significant level in multi group comparisons. In agreement with our data, several studies in mice utilizing forced oscillation technique have also found no significant differences after infection with RSV A2 [12, 16, 26]. Histologically, inflammation associated with RSV peaked 5 days post-infection in the cotton rat lungs, and inflammation was mild with predominately lymphocytic peribronchiolar cuffs as is consistent with previous studies [32]. The mild inflammation associated with RSV is also consistent with the lack of clinical signs and weight loss, and lack of perturbations in the lung function measurements. In contrast to mice, where a two-fold increase in mucus production [16] has been reported, no difference in mucus production was found after RSV infection. The only significant difference was the increase in pulmonary edema at 2 days post-infection. However, these studies did record higher baseline airway resistance in uninfected mice, as well as higher peak airway resistance values in uninfected mice, RSV infected mice, and mice sensitized with an allergen (ovalbumin) possibly due to the smaller airway diameters in mice [12, 16, 18, 40, 42, 43, 51, 52]. HDM sensitized cotton rats had significantly higher histologic inflammatory scores with a distinct eosinophilic component as observed previously [36], as well as inflammatory infiltrates at all locations (peribronchiole, perivascular, alveolar spaces and septae) in contrast to the predominantly peribronchiolar infiltrates of RSV infected cotton rats. The more severe inflammation associated with HDM sensitization was also associated with perturbations in lung function. There was a significant difference in lung mechanics after HDM sensitization.

These alterations in pulmonary function correlate with the higher inflammatory scores, suggesting that the severity of respiratory dysfunction is related to inflammation.

In summary, the effect of RSV infection in the cotton rat has been evaluated using state of the art technology. Consistent with the low level of inflammation and mucus production seen in RSV infected cotton rats, disturbances of lung function measured were not significantly different from uninfected cotton rats. Our data indicate, however, that lung function analysis in the cotton rat is valuable for other respiratory diseases, such as asthma or other viral infections that induce more severe inflammation.

## Supporting information

**S1 Fig. Mucus production in female uninfected cotton rats and cotton rats 5 days post-infection with RSV A2.** Quantification of PAS and Alcian blue positive (mucus) staining in bronchioles was performed using a semi-quantitative mucus scoring system. The mean and standard deviations are represented (n = 4/ group). There was no significant difference between groups (unpaired student T test, p value > 0.05).
(TIF)

**S2 Fig. Pulmonary edema after RSV A2 infection in female cotton rats.** The lung wet: dry weigh ratios in uninfected cotton rats and cotton rats 2, 4 or 6 days post-RSV infection. Mean wet: dry weight ratios and standard deviations are represented. There is no significant difference when all groups were compared to one another by One-Way ANOVA, p > 0.05; uninfected (n = 3), 2DPI RSV (n = 2), 4DPI RSV (n = 4), 6 DPI RSV (n = 4).
(TIF)

**S3 Fig. Comparison of male versus female bronchiolar widths.** The diameter of an average of 43 small airways (bronchioles) per uninfected animal were measured microscopically. The mean and standard deviation for each group are represented. One-way ANOVA, p>0.05 (n = 4).
(TIF)

**S4 Fig. Lung mechanics curves in uninfected cotton rats and 2, 4, 6 and 28 days post-RSV A2 infection female cotton rats.** The mean and standard deviation for each group are represented. Figure a represents total airway resistance (R). Figure b represents tissue damping (G). Figure c represents dynamic pulmonary compliance (C). Uninfected (n ranges 7–6), 2DPI RSV (n = 3), 4DPI RSV (n ranges 7–6), 6DPI RSV (n = 5), 28DPI RSV (n ranges 6–5).
(TIF)

**S5 Fig. Comparison of baseline compliance between uninfected and 4 days post-RSV infected cotton rats.** The mean and standard deviation for each group are represented. Both female and male cotton rats were presented in each group. Baseline pulmonary compliance was measured using forced oscillation technique. Asterisks indicate p< 0.05 by an unpaired two-tailed Student T-test.
(TIF)

**S6 Fig. Effect of system on lung function measurements.** Pulmonary function was measured in female cotton rats using the FV-M2 (old) and FX3 (new) flexivent system in uninfected and 4 days post-infection (DPI) with RSV in female cotton rats. The mean and standard deviation for each group are represented. Figures a-c represent total airway resistance (R). Figures d-f represent tissue damping (G). Figures g-i represent dynamic pulmonary compliance (C). Figures a, d, and g represent the lung mechanic curves for R, G, and C. Figures b, e, and h represent the baseline measurements for R, G, and C. Figures c, f, and i represent the measurements

obtained at 50mg/mL methacholine dosage for R, G, and C. Bars indicate $p < 0.05$ by One-way ANOVA; uninfected old system (n ranges 7–6), 4DPI RSV old system RSV (n ranges7-6), uninfected new system (n = 5), 4DPI RSV new system (n = 5).
(TIF)

**S7 Fig. Lung mechanics curves in uninfected cotton rats and HDM-sensitized female cotton rats.** The mean and standard deviation for each group are represented. A) Measurement of total airway resistance (R). B) Measurement of issue damping (G). C) Measurement of dynamic pulmonary compliance (C). Uninfected (n ranges 7–6), HDM (n = 4).
(TIF)

**S1 Table. Semi-quantitative histologic inflammatory scoring system.**
(DOCX)

**S2 Table. Semi-quantitative histologic mucus scoring system.**
(DOCX)

**S3 Table. Collar sensor clip pulse oximetry measurements.**
(DOCX)

**S4 Table. Thigh sensor clip pulse oximetry measurements.**
(DOCX)

**S5 Table. Arterial blood gas measurements.**
(DOCX)

## Acknowledgments

The authors would like to thank Dr. La Perle and the Comparative Pathology and Mouse Phenotyping Shared Resource at The Ohio State University for the help with making and scanning the histology slides for this project. The authors would also like to thank Tim Vojt for his help with formatting the images for publication.

## Author Contributions

**Conceptualization:** Stefan Niewiesk.

**Data curation:** Margaret E. Martinez, Olivia E. Harder.

**Formal analysis:** Margaret E. Martinez, Ian C. Davis.

**Funding acquisition:** Margaret E. Martinez, Stefan Niewiesk.

**Investigation:** Margaret E. Martinez, Stefan Niewiesk.

**Methodology:** Margaret E. Martinez, Olivia E. Harder, Lucia E. Rosas, Lisa Joseph, Ian C. Davis.

**Project administration:** Margaret E. Martinez, Stefan Niewiesk.

**Resources:** Ian C. Davis.

**Supervision:** Ian C. Davis, Stefan Niewiesk.

**Validation:** Margaret E. Martinez, Olivia E. Harder, Lucia E. Rosas, Ian C. Davis.

**Visualization:** Stefan Niewiesk.

**Writing – original draft:** Margaret E. Martinez.

Writing – review & editing: Margaret E. Martinez, Olivia E. Harder, Lucia E. Rosas, Ian C. Davis, Stefan Niewiesk.

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
