## [Decision Letter · Decision Letter 0]

3 Jun 2020

PONE-D-20-10846

Pulmonary function analysis in cotton rats after respiratory syncytial virus infection

PLOS ONE

Dear Dr. Martinez,

Thank you for submitting your manuscript to PLOS ONE. After careful consideration, we feel that it has merit but does not fully meet PLOS ONE’s publication criteria as it currently stands. Therefore, we invite you to submit a revised version of the manuscript that addresses the points raised during the review process.

Both reviewers raised important questions that need to be addressed concerning the choice of viral strains and the timing of the measurements for some of the lung function parameters.  In addition, several clarifications were requested regarding the experimental design and the description of the methods. Please respond to each of the criticisms raised by the reviewers. In some cases it may be necessary to perform additional experiments in order to fully address a reviewer concern.

We look forward to receiving your revised manuscript.

Kind regards,

Steven M. Varga, Ph.D.

Academic Editor

PLOS ONE

Journal Requirements:

'M.E.M. was funded through the Genetech Fellowship under grant number GRT00044407 from Genentech (https://www.gene.com/). O.E.H. and S.N. were funded through grant number P01AI112524 from the National Institute of Health National Institute of Allergy and Infectious Diseases (https://www.niaid.nih.gov/). L.E.R., L. J. and I.C.D. were funded through grant number HL137090 from National Institutes of Health/National Heart Lung & Blood Institute (https://www.nhlbi.nih.gov/).

The funders had no role in study design, data collection and analysis, decision to publish, or preparation of the manuscript.'

We note that you received funding from a commercial source: Genentech

5. Please include captions for your Supporting Information files at the end of your manuscript, and update any in-text citations to match accordingly. Please see our Supporting Information guidelines for more information: http://journals.plos.org/plosone/s/supporting-information

Reviewers' comments:

Reviewer's Responses to Questions

**Comments to the Author**

1. Is the manuscript technically sound, and do the data support the conclusions?

Reviewer #1: Partly

Reviewer #2: Yes

2. Has the statistical analysis been performed appropriately and rigorously? 

Reviewer #1: Yes

Reviewer #2: Yes

3. Have the authors made all data underlying the findings in their manuscript fully available?

Reviewer #1: Yes

Reviewer #2: Yes

4. Is the manuscript presented in an intelligible fashion and written in standard English?

Reviewer #1: Yes

Reviewer #2: Yes

5. Review Comments to the Author

Reviewer #1: The manuscript by Martinez et al describes an interesting work on forced oscillation technique in cotton rats infected with RSV or sensitized with HDM. This work is important, as application of this technique to cotton rats is novel and multiple parameters need to be established for the model. The use of HDM sensitization to achieve this task is justified and appeared to be highly successful. The application of technique in RSV-infected cotton rats, however, yielded some conflicting results. It was reported by one study that RSV infection in female cotton rats led to increased airway reactivity on day 4 pi (Fig.4a), while the data apparently could not be reproduced by another study (Fig.5a). Since this observation could be one of the most important conclusions of this work, a question arises as to whether suboptimal conditions were chosen for RSV infection. RSV infectious dose of 105 TCID50 per animal was chosen, yet this dose is associated with minimal inflammation in the cotton rat lungs, as was confirmed by data shown in Fig. 1 of this manuscript. Increasing RSV challenge 5- to 10-fold from the amount chosen by the authors of this manuscript is known to significantly increase pulmonary inflammation in cotton rats, and could be a more appropriate model for assessing airway reactivity in the model. Additional, it is not clear why RSV-line 19F was chosen for some studies along with the RSV A2 (e.g., mucus production), but not other studies where RSV A2 strain was used alone (e.g., edema, airway hyperreacticity). It appears that inclusion of both strains in all types of assessment, and using both viruses at a higher challenge dose could have increased informative potential of this work as it applies to RSV infection. Additionally, the selection of some other aspects of methodology is also not clear. For example, mucus production was assessed on d8 post-infection, with the date selection based on previous murine studies, as indicated by the authors. Yet, dynamics of RSV infection in cotton rats and mice is different, therefore d8 assessment could be suboptimal for mucus production assessment in RSV-infected cotton rats. Moreover, the sex differences were addressed in some studies (e.g., airway hyperreactivity), and not others (e.g., mucus production, edema), with justification for this choice not being clear. Overall, this work addresses an important issue and shows application of a new technique in cotton rats, demonstrating difference as they apply to HDM sensitization. The conditions for RSV infection, however, may need to be expanded to more accurately assess pulmonary function in the model.

Misc. comments:

-Fig.1 legend: description does not appear to accurately correspond to specific panels.

-There is no description of horizontal bars

-BAL WBS count is shown, but no method is described in the Materials and Methods.

-Inflation of the entire cotton rat lung (for the animal of indicated age) with 1 ml 4% paraformaldehyde can result in under inflation. Please check the amount used and reported (p.5, line 103).

-Table on p.6-8 may be redundant considering the reference cited and could be removed.

-It is not clear if all of the experiments were repeated and if so, how reproducible the results have been.

Reviewer #2: In this manuscript, the authors measured lung inflammation/mucus, arterial blood gas, and lung physiology in the cotton rat model of respiratory syncytial virus infection. They found that RSV did not elicit substantial lung/airway inflammation at days 4/5 after infection, did not significantly change arterial blood gas, and only caused airways responsiveness to methacholine challenge in female mice. In contrast, cotton rats that had been sensitized and challenged to house dust mite had significant increases in airway inflammation, mucus and airways responsiveness. The manuscript is well written, and the results presented are clear and convincing. Given that this is the first report of lung mechanics in the cotton rat in the setting RSV-induced disease, the results are useful in regard to vaccine and other approaches to treatment in future studies. It would have also been interesting to see if RSV infection during the course of allergic airway inflammation, such as in the dust mite model the authors use, would have caused exacerbated airways responsiveness as is seen in humans with asthma or in the mouse model of combined allergic and RSV disease.

Major points

Line 217 - There are concerns with the timing of the measurements of endpoints. For instance, as the authors point out, RSV line 19 elicited airway mucus and methacholine-induced airway responsiveness in mice, yet this occurred at day 8 after infection, after the appearance of the adaptive immune response and the activation of ILC2. Therefore, the absence of these findings in this report may have been related to measurements at times when IL-13 was not present, rather than the cotton rat not having either mucus or airways responsiveness. The lack of an extended time course of measurement of endpoints at the time of the peak of the adaptive immune response should be noted in the discussion. Of course, it is certainly possible that RSV line 19 or other strains that cause airway mucus expression and airways responsiveness in mice might not cause such findings in the cotton rat, but it is curious why the authors performed the measurements at the times that they did.

Line 219 - It is unclear why the authors assessed BAL cell counts on day 4/5 before the adaptive immune response has had time to develop. This likely reflects the low level of airway inflammation that they see at this time point. Openshaw published that innate cells such as NK cells and neutrophils are present within the first 4 days after infection, but then CD8+ and CD4+ cells are more abundant at days 7 through 9 after infection (Clin Mircobiol Rev 18:541-555, 2005)

Minor point

Line 141 - The authors state that the cotton rats were anesthetized for FlexiVent measurements and that the blood was obtained from carotid artery for measurement of arterial blood gas. The timing of the blood sampling relative to the pulmonary function measurements is not stated and is very important. If the methacholine challenge was performed prior to ABG measurement, then the bronchoconstriction from that procedure may have lowered the ABG measurement as methacholine causes airway obstruction. If on the other hand, the ABG was performed first, then the blood loss may have confounded the methacholine challenge measurement. I am not recommending that the experiments be performed again but want to make sure it is clear when the ABG measurement was made relative to the methacholine challenge. Also, the possibility of confounding the measurements based on the timing should be mentioned in the discussion.

6. PLOS authors have the option to publish the peer review history of their article (what does this mean?). If published, this will include your full peer review and any attached files.

Reviewer #1: No

Reviewer #2: Yes: Stokes Peebles

---

## [Author Response · Author response to Decision Letter 0]

7 Jul 2020

Responses to reviewers:

Reviewer #1: 

1. It was reported by one study that RSV infection in female cotton rats led to increased airway reactivity on day 4 pi (Fig.4a), while the data apparently could not be reproduced by another study (Fig.5a). Since this observation could be one of the most important conclusions of this work, a question arises as to whether suboptimal conditions were chosen for RSV infection. RSV infectious dose of 105 TCID50 per animal was chosen, yet this dose is associated with minimal inflammation in the cotton rat lungs, as was confirmed by data shown in Fig. 1 of this manuscript. Increasing RSV challenge 5- to 10-fold from the amount chosen by the authors of this manuscript is known to significantly increase pulmonary inflammation in cotton rats, and could be a more appropriate model for assessing airway reactivity in the model.

The difference between groups found in Figure 4a reflects the fact that a small number of groups was compared to each other, and therefore the statistical comparison detected a small difference. In Figure 5a, due to the comparison of a larger number of groups the small differences were not statistically significant. We concluded that the small difference in lung function was not reproducible nor clinically relevant which admittedly might be a conservative interpretation of the data.

In our hands, an inoculation dose of 105 TCID50 RSV is physiologically most relevant. We found that virus replication in lung tissue increased from 101 to 105 TCID50 [1] but did not further increase after inoculation of 106 TCID50 and inflammation as determined by BALF was also not significantly increased with a higher does [2]. A concern with the investigation of lung function after infection in a rodent model with a superphysiological dose is the induction of an artificially high level of inflammation due to a large amount of virus particles rather than virus replication. An example of this was published by us using a superphysiologic dose of measles virus (107 pfu instead of 105 pfu) [3].

2. I it is not clear why RSV-line 19F was chosen for some studies along with the RSV A2 (e.g., mucus production), but not other studies where RSV A2 strain was used alone (e.g., edema, airway hyperreacticity). It appears that inclusion of both strains in all types of assessment, and using both viruses at a higher challenge dose could have increased informative potential of this work as it applies to RSV infection. 

In cotton rats, RSV line 19F replicates and induces inflammation like RSV A2 (the commonly used virus strain in cotton rats). Previous studies in mice have found a difference in mucus production after infection with RSV line 19F vs RSV A2 [4, 5]. The determination of the ability of RSV line 19F to potentially induce mucus production in cotton rats was the only rational for using it (added in line 281-282). 

3. The selection of some other aspects of methodology is also not clear. For example, mucus production was assessed on d8 post-infection, with the date selection based on previous murine studies, as indicated by the authors. Yet, dynamics of RSV infection in cotton rats and mice is different, therefore d8 assessment could be suboptimal for mucus production assessment in RSV-infected cotton rats. 

In total we assessed the effect of RSV infection in cotton rats by histology at day 2, 4, 5, 6, 8, and 28 post-infection. Some studies have demonstrated that mucus production is related to neutrophilic enzymes [6-8] which correlate with peak inflammation. For this reason we have also assessed mucus production 5 days post-RSV infection using a semi-quantitative scoring scheme (see added S4 Table) but could not detect any increase in mucus production in cotton rat lungs.

4. The sex differences were addressed in some studies (e.g., airway hyperreactivity), and not others (e.g., mucus production, edema), with justification for this choice not being clear. 

We used both sexes for all measurements, and performed a gender-specific analysis. However, this is only reported in the figures if there is a gender specific difference. This information has been added to M&M (added line- 225-227). In addition, supplemental figures S2 and S5 have been added to expand upon the gender differences or lack thereof in cotton rats after RSV infection. 

Misc. comments:

-Fig.1 legend: description does not appear to accurately correspond to specific panels.

Changes have been made. 

-There is no description of horizontal bars

In the figure legends, it is stated that the bars indicate a significant difference (p<0.05). 

-BAL WBS count is shown, but no method is described in the Materials and Methods.

This has been added at line 131-138.

-Inflation of the entire cotton rat lung (for the animal of indicated age) with 1 ml 4% paraformaldehyde can result in under inflation. Please check the amount used and reported (p.5, line 103).

The cotton rat lung is relatively small compared to its body volume. We have tested different inoculation doses and found that greater than 1mL fluid induces histological artifacts that could be misconstrued as edema. 

-Table on p.6-8 may be redundant considering the reference cited and could be removed.

The table has been moved to supplemental materials (S1 Table). 

-It is not clear if all of the experiments were repeated and if so, how reproducible the results have been.

Individual experiments were repeated 3 to 5 times. Aggregate animal numbers were statistically analyzed (added in M&M, line 224)

Reviewer #2 

Major points

1. Line 217 - There are concerns with the timing of the measurements of endpoints. For instance, as the authors point out, RSV line 19 elicited airway mucus and methacholine-induced airway responsiveness in mice, yet this occurred at day 8 after infection, after the appearance of the adaptive immune response and the activation of ILC2. Therefore, the absence of these findings in this report may have been related to measurements at times when IL-13 was not present, rather than the cotton rat not having either mucus or airways responsiveness. The lack of an extended time course of measurement of endpoints at the time of the peak of the adaptive immune response should be noted in the discussion. Of course, it is certainly possible that RSV line 19 or other strains that cause airway mucus expression and airways responsiveness in mice might not cause such findings in the cotton rat, but it is curious why the authors performed the measurements at the times that they did.

The goal of our study was to determine the effects of acute RSV infection on lung function in cotton rats. This was based on the fact that many studies seem to correlate measurable lung dysfunction in laboratory rodents with increased inflammation and damage, such as models of acute respiratory distress syndrome [9, 10].In the cotton rat, adaptive immune responses correlate well with peak virus titers in lung tissue at day 4 and 6 [11, 12]. We did test later time points (day 28) based on previous publications demonstrating persistent inflammation thus providing ample time for an adaptive immune response to be mounted [13]. Many of the previous studies that suggest later time points for evaluating lung function were performed in mice and typically used whole body plethysmography or other less sensitive and accurate methods than forced oscillation technique [5, 14]. 

2. Line 219 - It is unclear why the authors assessed BAL cell counts on day 4/5 before the adaptive immune response has had time to develop. This likely reflects the low level of airway inflammation that they see at this time point. Openshaw published that innate cells such as NK cells and neutrophils are present within the first 4 days after infection, but then CD8+ and CD4+ cells are more abundant at days 7 through 9 after infection (Clin Microbiol Rev 18:541-555, 2005)

In the cotton rat, adaptive immune responses correlate well with peak virus titers in lung tissue at day 4 and 6 [11]. We assessed inflammation at days 4 and 5 via histology and/or BAL. Consistent with finding by others [12, 13] peak inflammation correlates with peak RSV replication and decreases from day 6 onward. 

Minor point

3. Line 141 - The authors state that the cotton rats were anesthetized for FlexiVent measurements and that the blood was obtained from carotid artery for measurement of arterial blood gas. The timing of the blood sampling relative to the pulmonary function measurements is not stated and is very important. If the methacholine challenge was performed prior to ABG measurement, then the bronchoconstriction from that procedure may have lowered the ABG measurement as methacholine causes airway obstruction. If on the other hand, the ABG was performed first, then the blood loss may have confounded the methacholine challenge measurement. I am not recommending that the experiments be performed again but want to make sure it is clear when the ABG measurement was made relative to the methacholine challenge. Also, the possibility of confounding the measurements based on the timing should be mentioned in the discussion.

We used separate groups of animals for the flexivent and the ABG measurements as both are terminal procedures. 

References: 

1. Wethington D, Harder O, Uppulury K, Stewart WCL, Chen P, King T, et al. Mathematical modelling identifies the role of adaptive immunity as a key controller of respiratory syncytial virus in cotton rats. Journal of the Royal Society, Interface. 2019;16(160):20190389. Epub 2019/11/28. doi: 10.1098/rsif.2019.0389. PubMed PMID: 31771450.

2. Green MG, Petroff N, La Perle KMD, Niewiesk S. Characterization of Cotton Rat (Sigmodon hispidus) Eosinophils, Including Their Response to Respiratory Syncytial Virus Infection. Comparative medicine. 2018;68(1):31-40. Epub 2018/02/21. PubMed PMID: 29460719; PubMed Central PMCID: PMCPMC5824137.

3. P. NSaG. Development of neutralizing antibodies correlates with resolution of interstitial pneumonia after measles virus infection in cotton rats. Journal of Experimental Animal Science. 1999;(40):201-10.

4. Moore ML, Chi MH, Luongo C, Lukacs NW, Polosukhin VV, Huckabee MM, et al. A chimeric A2 strain of respiratory syncytial virus (RSV) with the fusion protein of RSV strain line 19 exhibits enhanced viral load, mucus, and airway dysfunction. Journal of virology. 2009;83(9):4185-94. Epub 2009/02/13. doi: 10.1128/jvi.01853-08. PubMed PMID: 19211758; PubMed Central PMCID: PMCPMC2668460.

5. Boyoglu-Barnum S, Gaston KA, Todd SO, Boyoglu C, Chirkova T, Barnum TR, et al. A respiratory syncytial virus (RSV) anti-G protein F(ab')2 monoclonal antibody suppresses mucous production and breathing effort in RSV rA2-line19F-infected BALB/c mice. Journal of virology. 2013;87(20):10955-67. Epub 2013/07/26. doi: 10.1128/jvi.01164-13. PubMed PMID: 23885067; PubMed Central PMCID: PMCPmc3807296.

6. Boukhvalova MS, Prince GA, Blanco JC. The cotton rat model of respiratory viral infections. Biologicals : journal of the International Association of Biological Standardization. 2009;37(3):152-9. Epub 2009/04/28. doi: 10.1016/j.biologicals.2009.02.017. PubMed PMID: 19394861; PubMed Central PMCID: PMCPMC2882635.

7. Cortjens B, de Boer OJ, de Jong R, Antonis AF, Sabogal Pineros YS, Lutter R, et al. Neutrophil extracellular traps cause airway obstruction during respiratory syncytial virus disease. The Journal of pathology. 2016;238(3):401-11. Epub 2015/10/16. doi: 10.1002/path.4660. PubMed PMID: 26468056.

8. Stokes KL, Currier MG, Sakamoto K, Lee S, Collins PL, Plemper RK, et al. The respiratory syncytial virus fusion protein and neutrophils mediate the airway mucin response to pathogenic respiratory syncytial virus infection. Journal of virology. 2013;87(18):10070-82. Epub 2013/07/12. doi: 10.1128/jvi.01347-13. PubMed PMID: 23843644; PubMed Central PMCID: PMCPmc3753991.

9. Song W, Yu Z, Doran SF, Ambalavanan N, Steele C, Garantziotis S, et al. Respiratory syncytial virus infection increases chlorine-induced airway hyperresponsiveness. American journal of physiology Lung cellular and molecular physiology. 2015;309(3):L205-10. Epub 2015/06/14. doi: 10.1152/ajplung.00159.2015. PubMed PMID: 26071553; PubMed Central PMCID: PMCPmc4525118.

10. Aeffner F, Bolon B, Davis IC. Mouse Models of Acute Respiratory Distress Syndrome: A Review of Analytical Approaches, Pathologic Features, and Common Measurements. Toxicologic pathology. 2015;43(8):1074-92. Epub 2015/08/25. doi: 10.1177/0192623315598399. PubMed PMID: 26296628.

11. Rajagopala SV, Singh H, Patel MC, Wang W, Tan Y, Shilts MH, et al. Cotton rat lung transcriptome reveals host immune response to Respiratory Syncytial Virus infection. Scientific reports. 2018;8(1):11318. Epub 2018/07/29. doi: 10.1038/s41598-018-29374-x. PubMed PMID: 30054492; PubMed Central PMCID: PMCPMC6063970 is a for profit contract research organization that uses the cotton rat model for pre-clinical studies.

12. Prince GA, Jenson AB, Horswood RL, Camargo E, Chanock RM. The pathogenesis of respiratory syncytial virus infection in cotton rats. The American journal of pathology. 1978;93(3):771-91. Epub 1978/12/01. PubMed PMID: 362946; PubMed Central PMCID: PMCPMC2018360.

13. Grieves JL, Yin Z, Durbin RK, Durbin JE. Acute and Chronic Airway Disease After Human Respiratory Syncytial Virus Infection in Cotton Rats (Sigmodon hispidus). Comparative medicine. 2015;65(4):315-26. Epub 2015/08/28. PubMed PMID: 26310461; PubMed Central PMCID: PMCPMC4549677.

14. Wu YH, Lai AC, Chi PY, Thio CL, Chen WY, Tsai CH, et al. Pulmonary IL-33 orchestrates innate immune cells to mediate RSV-evoked airway hyperreactivity and eosinophilia. Allergy. 2019. Epub 2019/10/18. doi: 10.1111/all.14091. PubMed PMID: 31622507.

---

## [Decision Letter · Decision Letter 1]

27 Jul 2020

Pulmonary function analysis in cotton rats after respiratory syncytial virus infection

PONE-D-20-10846R1

Dear Dr. Martinez,

We’re pleased to inform you that your manuscript has been judged scientifically suitable for publication and will be formally accepted for publication once it meets all outstanding technical requirements.

Kind regards,

Steven M. Varga, Ph.D.

Academic Editor

PLOS ONE

Additional Editor Comments (optional):

Reviewers' comments:

Reviewer's Responses to Questions

**Comments to the Author**

1. If the authors have adequately addressed your comments raised in a previous round of review and you feel that this manuscript is now acceptable for publication, you may indicate that here to bypass the “Comments to the Author” section, enter your conflict of interest statement in the “Confidential to Editor” section, and submit your "Accept" recommendation.

Reviewer #1: All comments have been addressed

Reviewer #2: All comments have been addressed

2. Is the manuscript technically sound, and do the data support the conclusions?

Reviewer #1: Yes

Reviewer #2: Yes

3. Has the statistical analysis been performed appropriately and rigorously? 

Reviewer #1: Yes

Reviewer #2: Yes

4. Have the authors made all data underlying the findings in their manuscript fully available?

Reviewer #1: Yes

Reviewer #2: Yes

5. Is the manuscript presented in an intelligible fashion and written in standard English?

Reviewer #1: Yes

Reviewer #2: Yes

6. Review Comments to the Author

Reviewer #1: All comments have been adequately addressed. No additional comments/concerns are raised. The manuscript is recommended for acceptance for publication.

Reviewer #2: The authors have appropriately responded to my critiques and I believe that the manuscript is now ready for publication in PLoS One.

7. PLOS authors have the option to publish the peer review history of their article (what does this mean?). If published, this will include your full peer review and any attached files.

Reviewer #1: No

Reviewer #2: No

---

## [Editor Report · Acceptance letter]

30 Jul 2020

PONE-D-20-10846R1 

Pulmonary function analysis in cotton rats after respiratory syncytial virus infection 

Dear Dr. Martinez:

I'm pleased to inform you that your manuscript has been deemed suitable for publication in PLOS ONE. Congratulations! Your manuscript is now with our production department. 

Kind regards, 

on behalf of

Dr. Steven M. Varga 

Academic Editor

PLOS ONE